# Angiogenesis in Lung Cancer: Understanding the Roles of Growth Factors

**DOI:** 10.3390/cancers15184648

**Published:** 2023-09-20

**Authors:** Tchawe Yvan Sinclair Ngaha, Angelina V. Zhilenkova, Freddy Elad Essogmo, Ikenna K. Uchendu, Moses Owoicho Abah, Lionel Tabola Fossa, Zaiana D. Sangadzhieva, Varvara D. Sanikovich, Alexander S. Rusanov, Yuliya N. Pirogova, Alexander Boroda, Alexander Rozhkov, Jean D. Kemfang Ngowa, Leonid N. Bagmet, Marina I. Sekacheva

**Affiliations:** 1Institute for Personalized Oncology, Center for Digital Biodesign and Personalized Healthcare, First Moscow State Medical University of the Ministry of Health of Russia (Sechenov University), 8-2 Trubetskaya str., Moscow 119991, Russia; ngakhatchave_i@student.sechenov.ru (T.Y.S.N.); zhilenkova_a_v@staff.sechenov.ru (A.V.Z.); essogmo_f@student.sechenov.ru (F.E.E.); uchendu_i@student.sechenov.ru (I.K.U.); abakh_m@student.sechenov.ru (M.O.A.); sangadzhieva_z_d@staff.sechenov.ru (Z.D.S.); sanikovich_v_d@staff.sechenov.ru (V.D.S.); alexrus146@yandex.ru (A.S.R.); pirogova_yuliya96@mail.ru (Y.N.P.); alexcom90@yandex.ru (A.B.); alex009_97@mail.ru (A.R.); bagmetln@mail.ru (L.N.B.); 2Department of Public Health, James Lind Institute, Rue de la Cité 1, 1204 Geneva, Switzerland; 3Medical Laboratory Science Department, Faculty of Health Science and Technology, College of Medicine, University of Nigeria, Enugu Campus, Enugu 410001, Nigeria; 4Department of Oncology, Bafoussam Regional Hospital, Bafoussam 980, Cameroon; magnantr@yahoo.fr; 5Faculty of Medicine and Biomedical Sciences, University of Yaounde I, Yaounde 1364, Cameroon; jdkemfang@yahoo.fr

**Keywords:** angiogenesis, growth factors, lung cancer, tumor microenvironment, anti-angiogenic therapy

## Abstract

**Simple Summary:**

Growth factors promote angiogenesis, which is a critical process in the development of tumors. One of the therapeutic techniques being investigated in the treatment of cancer is the inhibition of angiogenesis through the inhibition of growth factors. This article aims to summarize the mechanisms by which growth factors influence the unfavorable evolution of lung cancers via angiogenesis as well as the therapeutic approaches that have been developed or are currently being developed in order to provide a foundation for researchers to investigate this question further and for practitioners to discuss therapeutic strategies when confronted with a lung cancer patient.

**Abstract:**

Research has shown the role of growth factors in lung cancer angiogenesis. Angiogenesis promotes lung cancer progression by stimulating tumor growth, enhancing tumor invasion, contributing to metastasis, and modifying immune system responses within the tumor microenvironment. As a result, new treatment techniques based on the anti-angiogenic characteristics of compounds have been developed. These compounds selectively block the growth factors themselves, their receptors, or the downstream signaling pathways activated by these growth factors. The EGF and VEGF families are the primary targets in this approach, and several studies are being conducted to propose anti-angiogenic drugs that are increasingly suitable for the treatment of lung cancer, either as monotherapy or as combined therapy. The efficacy of the results are encouraging, but caution must be placed on the higher risk of toxicity, outlining the importance of personalized follow-up in the management of these patients.

## 1. Introduction

With an estimated 2.2 million new cases and 1.8 million deaths, lung cancer was the second most commonly diagnosed cancer and the leading cause of cancer death in 2020 according to GLOBOCAN [1]. This cancer is broadly classified into two main types: the Non-Small Cell Lung Cancer (NSCLC) accounting for approximately 85% of cases, and the Small Cell Lung Cancer (SCLC), which is the most aggressive type, counts for about 15% of cases [2,3,4]. Despite the advancement of numerous therapeutic modalities, such as surgery, radiation therapy, chemotherapy, immunotherapy, and targeted therapy, the last two decades have been marked by a relatively low survival rate ranging from 10% to 20% in most countries, making lung cancer one of the deadliest cancers and a public health concern [1,5]. To address this issue, numerous treatment possibilities are being investigated, including those related to angiogenesis.

Angiogenesis is the development of new blood vessels that originate from already existing vasculature and is an important process in both normal and pathological conditions [6]. It has a significant impact in the progression and spread of cancers, particularly lung cancer [7,8]. Angiogenesis has been investigated for several years in several aspects, most notably its morphological characteristics, as determined by MRI, and its biological aspects, as determined by biomarker assays [9,10]. Researchers are interested in this phenomenon because it plays a substantial role in all tumoral processes and represents a new therapeutic avenue, as various compounds with anti-angiogenic properties are presently proposed as cancer treatments [7,11,12,13]. Many studies have proven the critical roles of Growth Factors in angiogenesis; nevertheless, the molecular processes underlying these roles, as well as how these mechanisms might be targeted in lung cancer therapy, are still to be fully understood. Furthermore, despite promising and encouraging results, medications targeting angiogenesis have only had limited clinical success in lung cancer treatment highlighting the need for a deeper understanding of this phenomenon and the therapeutic opportunities that arise from inhibiting the effects of Growth Factors [14,15,16,17].

## 2. Angiogenesis Affects the Lung Cancer Pattern through Several Mechanisms

Angiogenesis is essential in the development of lung cancer because it allows oxygen and nutrients to flow to rapidly developing tumor cells, contributing to tumor invasion and dissemination [7,18,19]. Angiogenesis may affect lung cancer pattern with a variety of methods.

### 2.1. Tumor Growth Enhancement

According to various study findings, angiogenesis provides cancer cells with an essential supply of oxygen and nutrients, aiding their rapid growth, and has been associated with an increase in tumor volume and a higher tumor grade in lung cancer [19,20]. This mechanism has been linked with numerous major angiogenic factors, including fibroblast growth factor 2 (FGF-2), epidermal growth factor (EGF), and vascular endothelial growth factor (VEGF). Indeed, the former growth factor promotes angiogenesis in lung cancer by stimulating endothelial cell migration and proliferation, which leads to the development of new blood vessels that support tumor growth [21,22].

### 2.2. Metastasis Promotion

Many studies agree that angiogenesis has a significant impact on tumorigenesis and metastatic processes by allowing tumor cells to spread through the creation of new blood circulation paths. Thus, the greater the neovascularization process, the greater the chance of metastasis. This indicates that there is a correlation between the amount of blood vessels that develop around the tumor and the metastatic potential of lung cancer [7,19,23]. Angiogenic factors including VEGF, FGF-2, and hypoxia-inducible factors (HIF) can also directly stimulate tumor cell proliferation, migration, and invasion [21,23,24,25,26].

### 2.3. Changes in Immunological Response in the Microenvironment

Indeed, angiogenesis can influence the immune response by promoting inflammation, which causes the release of pro-inflammatory cytokines, which promote cell survival and proliferation. These cytokines will also interact with immune cells, such as neutrophils and macrophages, encouraging the recruitment and infiltration of immune cells while also activating signaling pathways that promote angiogenesis. As a result, a vicious circle is formed, in which angiogenesis maintains the inflammatory process and the inflammatory process maintains angiogenesis [27,28]. Furthermore, these immune cells may help to build new blood vessels via a process known as vasculogenesis [19,29].

In addition to the aforementioned factors, new research has highlighted the contribution of epigenetic alterations in lung cancer angiogenesis, specifically how microRNAs influence pro- and anti-angiogenic factors [30,31,32]. During the typical immune response, the system generates a kind of self-tolerance that prevents immune cells from attacking indiscriminately via immune checkpoints. Tumor cells will stimulate checkpoint targets to protect themselves from being attacked in cancer. A preclinical trial combining angiogenesis inhibitors and immune checkpoint inhibitors produced promising results in targeting the intricate interplay between angiogenesis and immunological responses in lung cancer, demonstrating the potential for combination therapy to improve patient outcomes [31,33,34,35]. All of this emphasizes the significance of this research, which aims to reassess growth factors in order to better understand their role in the spectrum of lung cancer angiogenesis.

## 3. The Epidermal Growth Factor (EGF) Family, Their Receptors, and the Downstream

The EGF family is a group of glycoproteins that play roles in cell growth, survival, proliferation, and differentiation. These molecules are distinguished by a structural feature known as the “EGF-like domain,” which differentiates them from the others. This domain is in charge of the binding and activation of EGF receptors (EGFR) or similar receptors. Transforming growth factor alpha (TGF-alpha), amphiregulin (AREG), heparin-binding EGF-like growth factor (HB-EGF), beta-cellulin (BTC), epiregulin (EREG), and epigene (EPGN) are all members of the EGF family. Although each of these proteins has a unique biological function, they all have the ability to interact with EGFR to activate the downstream signaling pathways that drive cell proliferation, migration, and invasion [36,37,38].

In a number of malignancies, including lung cancer, EGFR have been shown to be found in very high concentrations and also play an important role in angiogenesis, tumor formation, and progression. Indeed, members of the EGF family are especially prevalent in lung cancer, not only because of the cancerous inflammatory process, but also because these patients are vulnerable to external attacks by various pathogens, which maintain the inflammatory reaction and thus constantly stimulate EGF production. All these thereby promote the occurrence of EGF mutations [39,40,41]. When EGFR are activated, multiple downstream signaling pathways are activated, including the PI3K-Akt and MAPK-ERK pathways, which have been found to stimulate the production of pro-angiogenic factors, such as VEGF and bFGF (basic Fibroblast Growth Factor). Furthermore, stimulation of EGFR can activate downstream transcription factors, such as hypoxia-induced factor 1 (HIF-1), which will stimulate the production of VEGF genes, genes that will trigger the synthesis of pro-angiogenic VEGF molecules via the translation process. Pro-angiogenic molecules, such as VEGF, bFGF, and HIF-1, are thus boosted through these several processes, promoting the proliferation, migration, and formation of new blood vessels [25,42,43,44,45]. Figure 1 depicts the several pathways that will be activated upon receptor stimulation that ultimately contribute to the process of angiogenesis.

Gefitinib, the first approved EGFR-TKI (EGFR—Tyrosine Kinase Inhibitors), was shown to be effective in treating non-small cell lung cancer (NSCLC) patients with EGFR mutations, leading to improved progression-free survival and response rates. Since then, many generations have been developed, and the fourth generation is now on the pre-clinical stage of development and aims to solve not only the problem of mutation of EGFR, but also resistance to the drugs of the third generation (Rociletinib and Osimertinib) [39,47].

Recent laboratory and clinical studies, such as that of Nakagawa et al. and that of Subbiah et al., have revealed new targets and therapeutic treatments centered on EGF family members and their influence on lung cancer angiogenesis [48,49,50]. In 2022, a study published by Nakagawa et al. found that targeting the EGF receptor in conjunction with anti-VEGF medication results in improved survival rates in lung cancer patients, pinpointing the potential advantage of targeting several pro-angiogenic pathways [48]. However, this should be approached with caution because certain studies have identified the possibility of toxicity leading to therapeutic termination [48,51]. Other studies also found that the EGF-like domain of HB-EGF increases angiogenesis in lung cancer by stimulating VEGF production and then driving endothelial cell migration and neovascularization [21,52].

In a clinical trial conducted by Rosell in 2017, the combination of the EGFR inhibitor erlotinib and the VEGFR inhibitor bevacizumab led to an increase in survival in individuals having advanced NSCLC, probably due to the synergistic effects of blocking both the EGF and VEGF pro-angiogenic pathways [53]. Other clinical trials have focused on the development of new EGF/EGFR targeted therapies, such as monoclonal antibodies and TKI, which may provide patients with lung cancer with better therapeutic alternatives [54,55].

Many clinical trials have primarily targeted the EGF pathway alone or in conjunction with other neighboring pathways, the leader of which is VEGF. Table 1 is a collection of some intriguing research on lung cancer, gathered from freely accessible American and European databases that cover many studies, among which clinical trials have been conducted or are in progress around the world [56,57].

## 4. Vascular Endothelial Growth Factor (VEGF)

VEGF are growth factors that stimulate the formation of new blood vessels from pre-existing ones. Several VEGF family members have been identified, each with distinct and partially overlapping activities [58,59].

Today, we know that the VEGF family consists of six separate members, which are VEGF-A, VEGF-B, VEGF-C, VEGF-D, VEGF-E, and placental growth factor (PlGF). The different members differ in their binding affinity and specificity to the three receptors VEGFR1, VEGFR2, and VEGFR3. The most intensively researched member of the family, VEGF-A, has been found to be a powerful proangiogenic agent. It is expressed by a wide range of cells, including tumor cells, and binds to its endothelial cell receptors, VEGFR1 and VEGFR2, to drive blood vessel growth [58,60].

The mechanism of action of VEGF in angiogenesis is complex and involves the interaction of VEGF with its receptors on endothelial cells. The interaction triggers a series of events that promote endothelial cells to multiply, relocate, and to survive, ultimately leading to the formation of new blood vessels [60,61,62]. VEGF also promotes the formation of the extracellular matrix and the recruitment of pericytes, both of which help to stabilize newly created vessels [63,64].

The overexpression of VEGF, frequently seen in lung cancer, is associated with enhanced angiogenesis, tumor development, and metastasis. Anti-VEGF medication bevacizumab has been approved as one of the possible choice to treat advanced cases of NSCLC [65]. Anti-VEGF medication improves progression-free survival and overall survival in people with advanced NSCLC, according to several trials. Novel anti-angiogenic drugs and combination therapies that target several pathways involved in tumor angiogenesis have recently been investigated in the field of angiogenesis in lung cancer. In a phase II clinical trial lead by Horn L. et al., for example, the VEGFR inhibitor bevacizumab was added to etoposide and cisplatin and used as a first-line therapy for people with advanced stage small cell lung cancer (SCLC), and this resulted in an increase in the survival rate compared to historical controls who received this chemotherapy regimen without bevacizumab [66]. Furthermore, combining the EGFR inhibitor, erlotinib, with the VEGFR inhibitor, bevacizumab, has also been shown to improve survival rates in advanced cases of non-small cell lung cancer [53]. For more information, see Table 1.

## 5. Colony Stimulating Factors (CSF)

CSF are a group of cytokines that regulate the production, differentiation, and function of white blood cells. They have been demonstrated to play a role in angiogenesis in addition to their involvement in regulating white blood cell formation and differentiation. [67,68]. There are four canonical members in the family, including [69]:-Granulocyte colony-stimulating factor (G-CSF): A cytokine that promotes the creation and development of neutrophils, a kind of white blood cell, from bone marrow progenitor cells;-Granulocyte-macrophage colony-stimulating factor (GM-CSF): A cytokine that stimulates the development and differentiation of bone marrow progenitor cells into neutrophils, monocytes, and macrophages;-Macrophage colony-stimulating factor (M-CSF): A cytokine that induces the production and maturation of macrophages from bone marrow progenitor cells;-Interleukin 3 (IL-3 or multi-CSF): A hematopoietic cytokine and colony-stimulating factor that aids in the growth and maturation of erythroid, myeloid, megakaryocyte, and lymphoid progenitors.

The exact mechanism by which G-CSFs regulate angiogenesis is not fully understood, but it is thought to involve the recruitment and activation of bone marrow-derived endothelial progenitor cells (EPCs). EPCs are cells that help to generate new blood vessels from circulating endothelial progenitors during postnatal vasculogenesis. G-CSFs have the ability to mobilize EPCs from the bone marrow and boost their differentiation, proliferation, and migration, resulting in increased angiogenesis [67,70,71]. However, the therapeutic implication of such a discovery remains very controversial since G-CSF is used in prophylaxis to avoid the febrile neutropenia often observed during chemotherapy, and this considerably reduces the interest in developing G-CSF inhibitors [72]. For additional information, see Table 1.

G-CSFs, especially G-CSF and GM-CSF, have been studied for their potential relevance in lung cancer. G-CSF levels were discovered to be associated with an unfavorable prognosis in cases of NSCLC. Other findings revealed that G-CSF and GM-CSF can accelerate tumor development and angiogenesis in lung cancer as well as inhibiting G-CSF signaling can diminish angiogenesis and tumor growth. [67,73,74,75]. Taking these factors into account, recent advances in the study of angiogenesis in lung cancer have focused on the possible use of G-CSFs as therapeutic targets [73].

IL-3 has not been widely explored in relation to lung cancer angiogenesis. However, there are some findings stating that it may play an essential part in inducing angiogenesis in other forms of cancer. IL-3 may increase cancer cell proliferation and survival via mechanisms such as tumor microenvironment modification and the activation of cell multiplication and sustainment signaling pathways [76,77]. More research is required to fully comprehend this cytokine’s potential role in angiogenesis and lung cancer progression.

## 6. Bone Morphogenetic Protein (BMP)

BMPs are a type of signaling molecule that belongs to the TGF-beta (transforming growth factor-beta) family. After being recognized for its ability to stimulate bone formation, BMPs were shown to have various additional activities, including influencing cell growth, differentiation, and death [78].

It has been demonstrated that BMPs have a complex and context-dependent role in angiogenesis. They can increase angiogenesis in specific circumstances by boosting endothelial cell differentiation, proliferation, and migration [79,80,81]. In other contexts, some evidence suggests that BMPs may decrease angiogenesis by increasing the expression of angiogenesis inhibitors, but the precise mechanism remains unknown [79,82,83].

BMPs have been identified as having a key role in lung cancer tumor angiogenesis and progression. BMPs, particularly BMP-2, BMP-4, and BMP-7, have been discovered to be elevated in lung cancer tissues and have been linked to a bad prognosis in these individuals. BMPs have also been demonstrated to induce the production of pro-angiogenic factors, resulting in angiogenesis stimulation and tumor growth in lung cancer [28,84,85,86]. As a result, recent research has focused on BMP signaling targeting as a viable therapeutic for lung cancer with substantial angiogenesis [87]. In 2021, Meng et al. suggest BMP5 as a potential crucial target for lung adenocarcinoma treatment [84].

## 7. Fibroblast Growth Factors 1 and 2 (FGF1 and FGF2)

FGF1 and FGF2 are potent angiogenic agents that increase vascular endothelial cell development, displacement, and survival. They attach to heparan sulfate proteoglycans on endothelial cell surfaces, activating both the FGFR1 and FGFR2 endothelial cell isoforms. This activates downstream signaling pathways, such as ERK, PI3K, and PLC, which promote angiogenesis by increasing the synthesis and secretion of pro-angiogenic molecules, such as VEGF and platelet-derived growth factor (PDGF) [88,89].

FGFs, especially FGF1 and FGF2, play a complex and context-dependent involvement role in lung cancer. FGFs have an important role in tumor angiogenesis and growth in early-stage lung cancer because they encourage the production of new blood vessels, which deliver nutrition and oxygen to the tumor cells [19,90,91]. FGF2 expression has been observed to be elevated in lung cancer and is associated with a bad prognosis [92]. FGFs can also contribute to anti-angiogenic therapy resistance since tumors can shift to an alternate angiogenic pathway that is not targeted by current therapies [19,93,94,95]. Figure 2 depicts an example where VEGFR is targeted and demonstrates the many compensatory angiogenic factors/signaling routes that tumors use to sustain the angiogenic process, with several growth factors, including FGF, being involved [96].

Recent research has focused on developing FGF signaling pathway inhibitors as potential lung cancer therapeutics. BGJ398, for example, is a selective inhibitor of the FGFR signaling pathway that has demonstrated potential anticancer effects in preclinical investigations and is currently being tested in clinical trials in patients with FGFR-mutant NSCLC [97,98].

## 8. Interleukins (IL)

Interleukins are a type of cytokine that is involved in immunological modulation as well as physiological processes, such as development, angiogenesis, and hematopoiesis. The interleukin family contains around 40 members, each with its own distinct function in the immune system [99,100].

Several interleukins, including IL-1 beta, IL-6, and IL-8, have been identified to be dysregulated in lung cancer. These interleukins have been demonstrated to stimulate tumor growth by angiogenesis as well as cell proliferation, survival, and invasion. For example, IL-6, IL-8, and IL-17 have been shown to increase VEGF expression in lung cancer cells, which increases angiogenesis [28,101,102,103].

Research has focused on targeting interleukins as a potential lung cancer therapy strategy [104,105]. Many researches have shown that IL-6 interacts with other molecules, notably VEGF to ultimately promote angiogenesis [106,107]. Figure 3 illustrates interleukin-6 in the tumor microenvironment, depicts how it interacts with other molecules as well as the VEGF pathway to promote the angiogenesis process and therefore favor the tumor progression, and also shows anti-IL-6 possible targeted molecules used in cancer therapy [106,108,109,110,111,112,113].

Researchers currently believe that certain interleukins have diagnostic and prognostic value when combined with other molecules. In 2022, a study lead by Yan X., for example, found that IL-6 and IL-8 could be used as possible molecular biomarkers to diagnose and predict lung cancer metastasis regardless of pathological type or to improve the specificity and sensitivity for the diagnosis of lung cancer when paired with Carcinoembryonic antigen (CEA) [114].

## 9. Others Growth Factors

-**Hepatocyte Growth Factor (HGF):** It is a cytokine with two different domains, one N-terminal and one C-terminal, each with its own set of biological activity. The C-terminal domain of HGF mediates its ability to induce angiogenesis by activating subsequent signaling pathways, such as the PI3K/Akt and MAPK/ERK pathways [115,116]. Its rise in lung cancer has been linked to a poor prognosis and resistance to anti-angiogenic therapy. Recent research found that an anti-HGF monoclonal antibody can inhibit HGF-induced angiogenesis and tumor growth in preclinical models of lung cancer, providing a potential therapeutic strategy for lung cancer patients [116,117].-**Human Epidermal Growth Factor Receptors 2 and 3 (HER2 and HER3):** These two belong to the family of tyrosine kinases receptors and are overexpressed or mutated in many cancers and increase angiogenesis by activating both the PI3K/Akt and MAPK/ERK signaling pathways [118]. Several HER2-targeting therapy treatments, including monoclonal antibodies and tyrosine kinase inhibitors (TKIs), such as afatinib and neratinib, have demonstrated success in preclinical and clinical trials. Moreover, many researchers are working to bring out new therapies targeting HER-2 in the field of lung cancer [119,120].-**Platelet Derived Growth Factor (PDGF) α/β**: They belong to the PDGF receptor tyrosine kinase family and have been linked to lung cancer angiogenesis. PDGFR-alpha and PDGFR-beta are both overexpressed in lung cancer, and their presence has been linked to a bad prognosis. In preclinical lung cancer models, blocking PDGF signaling has been shown to diminish tumor formation and angiogenesis [22,121]. As for the others, combination treatments targeting both the PDGF and VEGF signaling pathways in lung cancer have been examined. In one trial, the anti-PDGF agent nintedanib was coupled with the anti-VEGF agent bevacizumab in lung cancer patients, resulting in an improvement in progression-free survival when compared to bevacizumab alone [122].-**Soluble Tie 2 (sTie2)** is a shortened version of the Tie2 receptor, which is an angiopoietin receptor expressed on endothelial cells and is involved in angiogenesis and vascular stabilization [123]. Its expression has been linked to unfavorable outcomes in several malignancies, including lung cancer, and research is being conducted to see how it can be targeted for therapy [124].-**Soluble Neuropilin 1 (sNRP1)** is a shortened version of the neuropilin 1 receptor that is produced on endothelial cells and impacts angiogenesis by acting as a VEGF coreceptor [125]. As with soluble Tie 2, large levels of sNRP1 expression have been linked to a worse prognosis, and it is also a molecule of interest in the realm of targeted therapeutics for lung cancer [126,127].

## 10. Conclusions and Perspectives

Growth factors have a strong pro-angiogenic effect because they encourage the development of new vessels through many pathways, resulting in tumor progression and metastasis: That is why they constitute one of the therapeutic targets against cancers.

In lung cancer, in view of the rapid and unfavorable evolution sometimes observed, even in patients undergoing treatment, new therapeutic approaches have been proposed, and the first evaluations are encouraging.

Considering the complexity of the processes involved in angiogenesis and the multitude of growth factors that promote the therapeutic escape mechanism, combined therapies and therapies targeting the downstream signaling pathways are now being extensively explored as potentially of interest in the management of this disease. In the meantime, the literature remains favorable on the central role of EGFR-TKI-based treatment, even in the case of uncommon mutations.

A personalized approach with a prior analysis of genetic and molecular profiles in search of the presence of mutations in patients (EGFR mutations) is strongly recommended. Special attention should also be paid to the risk of toxicity when launching a therapeutic regimen because it constitutes one of the main complications of combined therapies and can thus justify the discontinuation of the treatment regimen.

Finally, we believe it is critical to emphasize the importance of continuing to conduct research in search of new lung cancer biomarkers, identifying all of the factors and mechanisms responsible for mutation occurrence, and deepening our understanding of the processes by which lung cancer cells develop resistance to anti-angiogenic therapies.

## Figures and Tables

**Figure 1 cancers-15-04648-f001:**
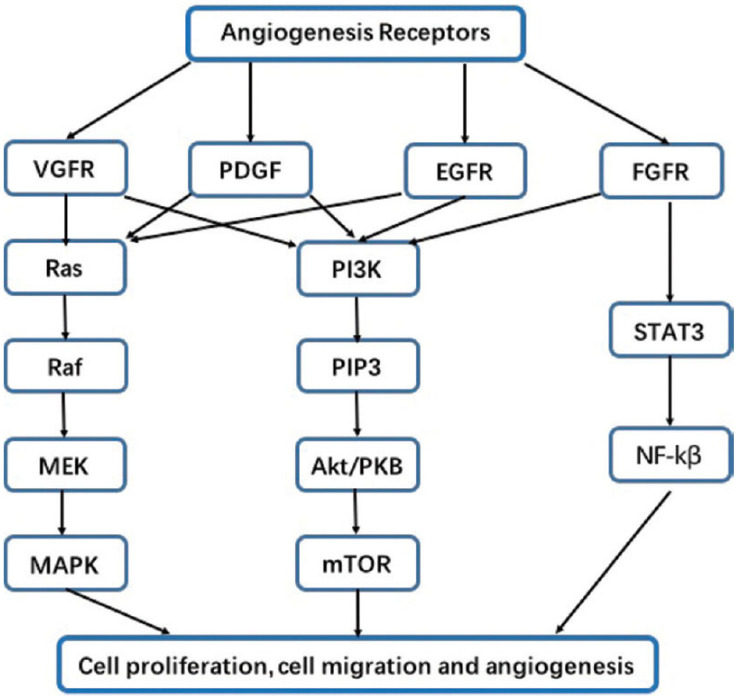
Associated receptors and their signaling pathways involved in angiogenesis. On the figure we can identify the different angiogenesis receptors and the pathways they stimulate. These stimulations ultimately lead to a variety of processes involved in angiogenesis, such as cell proliferation, migration, and survival. EGFR stimulates the RAS/RAF/ERK/MAPK (also called MAPK-ERK) and the PI3K/Akt signaling pathways. These pathways are also stimulated by VGFR, PDGF, and FGFR. The latter (FDFR) also triggers the STAT3/NF-κβ pathway. Data source: Review article “Molecular mechanisms involved in angiogenesis and potential target of anti-angiogenesis in human glioblastomas” by Xu Y, Yuan FE, Chen QX, and Liu BH [46] and published in an open access journal.

**Figure 2 cancers-15-04648-f002:**
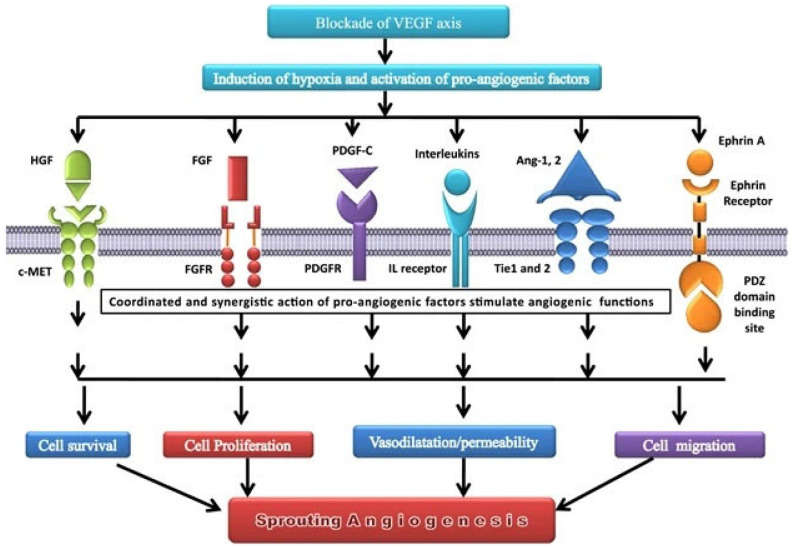
Compensatory angiogenic factors/signaling routes, including FGF, after blockade of VEGF axis. The inhibition of the VEGF axis causes tissue hypoxia, which the body interprets as a lack of vessels, preventing blood from reaching the appropriate areas. This activates pro-angiogenic factors, such as HGF, FGF, PDGF, interleukins, Ang-1&2, and ephrin A, resulting in a cascade of events leading to angiogenesis. Data source: Review article “Compensatory angiogenesis and tumor refractoriness” by Gacche [96], published in an open access journal.

**Figure 3 cancers-15-04648-f003:**
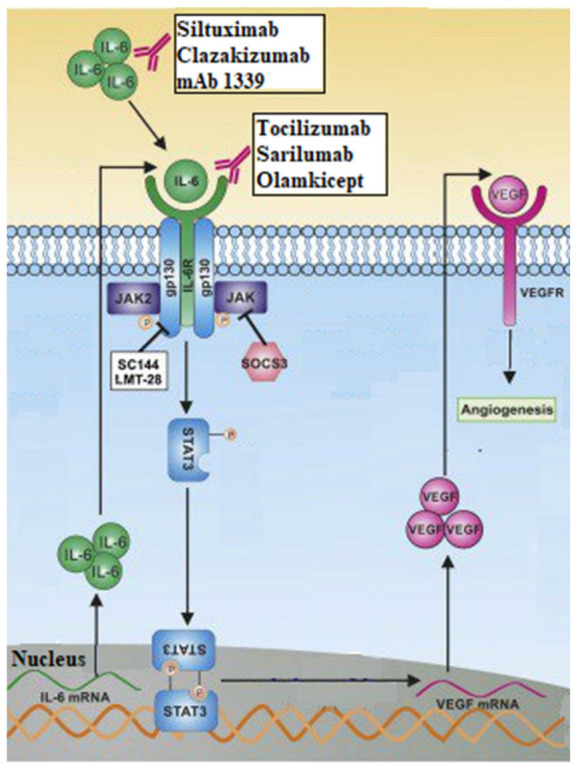
Interleukin-6 in the tumor microenvironment and its action on VEGF axis. The formation of IL-6 from stimulated IL-6 mRNA, leads to a release of IL-6, in high concentration, in the extracellular milieu. After connecting to its receptors, IL-6 initiates a chain of reactions that will ultimately lead to the synthesis of VEGF and therefore to angiogenesis. The figure also shows the different therapeutic molecules used to inhibit this pathway and therefore treat cancer. Data source: Adapted figure from the article “Cross-talk between EGFR and IL-6 drives oncogenic signaling and offers therapeutic opportunities in cancer” by Ray K, Ujvari B, Ramana V and Donald J. [113]. Re-use License number 5615250558075.

**Table 1 cancers-15-04648-t001:** Some additional researches in the subject of growth factor-based targeted therapy in lung cancer (data consulted on 6 July 2023) [56,57].

Last Update	Location and Study Identifier	Study Type	Study Title	Condition	Intervention	Status	Findings
May 2023	United Kingdom NCT04179890	Observational and retrospective	The study observes how long patients with non-small cell lung cancer (NSCLC) benefit from treatment with epidermal growth factor tyrosine kinase inhibitor (EGFR-TKI) when given either for uncommon mutations or for common mutations in the sequence afatinib followed by osimertinib (UpSwinG)	Non-squamous, Non-Small Cell Lung Cancer,	Observation of EGFR-TKI: -Afatinib -Erlotinib -Gefitinib -Osimertinib	Complete	treatment with EGFR-TKI should be considered as standard for most patients with uncommon mutations
February 2023	USA NCT05062980	Clinical Trial	Quaratusugene Ozeplasmid (Reqorsa) in combination with Pembrolizumab in previously treated Non-Small Cell Lung Cancer (Acclaim-2) Phase I/II	Non-Small Cell Lung Cancer	A: Quaratusugene ozeplasmid (pan-TKI: EGFR and Akt inhibitor) + Pembrolizumab (VEGFR downstream inhibitor: PD1 inhibitor) B: Docetaxel (microtubule inhibitor) + ramucirumab (VEGFR inhibitor) + 3rd molecule proposed by physician	On going	/
May 2019	United Kingdom NCT02109016	Clinical Trial	A single arm, open-label, phase II study to assess the efficacy of the dual VEGFR-FGFR tyrosine kinase inhibitor, Lucitanib, given orally as a single agent to patients with FGFR1-driven lung cancer.	Advance stage of Small and Non-small cell lung cancer with adenomatous, squamous, and large cell histologies, as well as FGF, VEGF, or PDGF genetic alterations.	Lucitanib, a VEGFR-FGFR tyrosine kinase inhibitor	Terminated	Interim analysis was either impossible (due to short time data collection) or showed low probability of clinically significant result
January 2013	USA NCT00862134	Clinical Trial	Randomized, Multi-center, Open-label, Study of PR104 Versus PR104/Docetaxel in Non-Small Cell Lung Cancer (NSCLC) Phase II	Non-Small Cell Lung Cancer	A: Docetaxel (microtubule inhibitor) B: Docetaxel + PR104 (hypoxia-activated prodrug) + G-CSF for prophylaxis	Terminated	Interim analysis indicated low probability of clinically significant result

## Data Availability

Not applicable.

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
