# Peer review of "Angiogenesis in Lung Cancer: Understanding the Roles of Growth Factors"

_cancers, 2023, doi:10.3390/cancers15184648_

Round 1

Reviewer 1 Report

This manuscript reviews the role of growth factors including EGF family, CSF family, BMPs, FGFs, IL-6, HGF, HER2, HER3, PDGFF sTie2 and sNRP1 in lung cancer angiogenesis. Tumor growth enhancement, metastasis promotion and changes in immunological response in the microenvironment are described as the mechanisms underlying the effect of angiogenesis on the lung cancer pattern. Authors provide the updated information regarding the role of these growth factors on the process of angiogenesis in lung cancer. The review is well written and organized. I recommend acceptance of this manuscript.

Author Response

Good morning dear reviewer and thank you very much for your encouraging feedback.

I remain open and available for any work collaboration in the field of cancer +/- public health. 

I wish you a happy end of week. 

Yours sincerely,

Dr Yvan NGAHA

Reviewer 2 Report

Tchawe and colleagues present a narrative review covering the biology of growth factors and associated mechanisms, promoting angiogenesis in lung cancer. The manuscript is clearly articulated and will serve as reference to understand the involvement of growth factors that are implicated in the induction of angiogenesis in lung cancer.

Specific comments

Introduction. A brief introduction of lung cancer including the most prevalent type NSCLC and SCLC (the data presented sorrounds these types) current treatment modalities and survival rate is essential.

Page 4, line 150-155. The authors stated that Nakagawa`s work was performed in lung cancer animal model (ref 46), which is incorrect. According to this reference, RELAY, ramucirumab plus erlotinib were tested in patients with untreated, metastatic, EGFR-mutated NSCLC. Ansari et al. 2022 is a review; in this case, authors should explicitly indicate the studies which reported the therapeutic interventions on EGF family members in patients.

Page 5, Table1, for better flow of thought, it is recommended to provide a separate section presenting the growth factor-based clinical studies.

In Table 1, for clearer overview, kindly revise the headings as follows:  Study Identifyer/status, Phase study, Trial design, Findings and Reference. Kindly use grids as necessary and define abbreviations at the bottom of the Table.

Figure 1. The description of the figure is insufficient. EGFR is not the only receptor that stimulates the RAS/RAF/ERK/MAPK and the PI3K/Akt pathways. Based on the diagram, VGFR, PDFR can also activate the former. In addition, VGFR. PDGF and FGFR can trigger the latter pathway. The STAT3 and NFκβ pathway was not mentioned at all.  

Figure 2. Other major biological factors in addition to FGF such as HGF, PDGF-C, interleukins and others should be included in the figure legends. Kindly define all abbreviations used.

Page 8, line 303-304. Include references of IL-6 targeted molecules used in cancer therapy including the inhibitors for gp130 and JAK as shown in Figure 3.  

Author Response

Dear reviewer,

Good morning and thank you for your feedback. With regard to the different points raised up, we plea you to find our answers in the following lines:

1- Introduction. A brief introduction of lung cancer including the most prevalent type NSCLC and SCLC (the data presented sorrounds these types) current treatment modalities and survival rate is essential. 

Answer: This has be done and an additional  paragraph has been added.

2- Page 4, line 150-155. The authors stated that Nakagawa`s work was performed in lung cancer animal model (ref 46), which is incorrect. According to this reference, RELAY, ramucirumab plus erlotinib were tested in patients with untreated, metastatic, EGFR-mutated NSCLC. Ansari et al. 2022 is a review; in this case, authors should explicitly indicate the studies which reported the therapeutic interventions on EGF family members in patients.

Answer: this part has been reviewed and amended.

3- Page 5, Table1, for better flow of thought, it is recommended to provide a separate section presenting the growth factor-based clinical studies.

Answer: we added an additional paragraph aiming to better explain and introduce the table. 

4- In Table 1, for clearer overview, kindly revise the headings as follows:  Study Identifyer/status, Phase study, Trial design, Findings and Reference. Kindly use grids as necessary and define abbreviations at the bottom of the Table.

Answer: we added an additional column aiming to present the study design. The phase of the clinical trials were already in the titles of the studies so, we found not necessary to state them again. Futhermore, apart from the problem related to  space used by the table, we couldn't add another column for reference since the studies are all contained in the data bases and are accessible only through their identifiers (provided in the second column) on the websites https://beta.clinicaltrials.gov/ and https://www.clinicaltrialsregister.eu/ctr-search/search 

5-Figure 1. The description of the figure is insufficient. EGFR is not the only receptor that stimulates the RAS/RAF/ERK/MAPK and the PI3K/Akt pathways. Based on the diagram, VGFR, PDFR can also activate the former. In addition, VGFR. PDGF and FGFR can trigger the latter pathway. The STAT3 and NFκβ pathway was not mentioned at all.  

Answer: We amended this part according to your proposals.

6-Figure 2. Other major biological factors in addition to FGF such as HGF, PDGF-C, interleukins and others should be included in the figure legends. Kindly define all abbreviations used.

Answer: We amended this part according to your proposals.

7- Page 8, line 303-304. Include references of IL-6 targeted molecules used in cancer therapy including the inhibitors for gp130 and JAK as shown in Figure 3. 

Answer: We amended this part according to your proposal and included the references related to the theurapeutic in the in the paragraph introducing our figure. 

We really thank you for your deep reviewing and we remain open to futher proposal to improve this work.  We also hope that these improvements will convince you to sign this review. 

We are open to any work collaboration on topic related to cancer and/or public health. 

thank you once more for all and please do have a nice week.

Best regards, 

Dr Yvan Sinclair NGAHA TCHAWE